# Perceptions of Health Risks and Accessibility: A Social Media-Based Pilot Study of Factors Influencing Use of Vaping and Combustible Tobacco Products

**DOI:** 10.3390/ijerph22050800

**Published:** 2025-05-20

**Authors:** Enitan Banjo, Zoya Ahadian, Nikita Kasaraneni, Howard Chang, Sarala Perera, Kristen Emory, Laura E. Crotty Alexander

**Affiliations:** 1Department of Health Sciences, California State University, Dominguez Hills (CSUDH), Carson, CA 90747, USA; zinat.banjo@gmail.com (E.B.); kemory@csudh.edu (K.E.); 2Department of Medicine, Division of Pulmonary, Critical Care, Sleep and Physiology, University of California San Diego (UCSD), San Diego, CA 92093, USA; zoyaahadian@gmail.com (Z.A.); nikita.kasaraneni@gmail.com (N.K.); chang12307@gmail.com (H.C.); s2perera@ucsd.edu (S.P.); 3Medicine Service, Pulmonary and Critical Care Section, Veterans Affairs San Diego Healthcare System, La Jolla, CA 92161, USA

**Keywords:** tobacco, e-cigarette vaping, cigarette smoking, multi-inhalant use, cessation, adolescents

## Abstract

The prevalence of e-cigarette use (vaping) in young adults is concerning, particularly because the initiation of combustible tobacco use is higher in e-cigarette vapers. It is unclear why young, never-smoker vapers decide to start smoking cigarettes, but they may be influenced by perceptions of health risks and accessibility. We designed a social media questionnaire to assess factors driving the initiation of combustible tobacco use by e-cigarette vapers (multi-inhalant use) and switches between inhalant types. Respondents reported an earlier initiation of combustible tobacco versus vaping (18 vs. 19, respectively, *p* < 0.0001), greater ease of obtaining combustibles versus vaping products (*p* < 0.01), and pleasure of vaping being equivalent to that of smoking. The majority of subjects (57%) reported smoking first prior to adding vaping devices, 32% reported initiating both smoking and vaping within 12 months of one another, and 11% reported initiating vaping first. Among respondents (n = 864) who switched from vaping to smoking (n = 104), the primary reasons included 1. the perception that smoking was healthier (44%) and 2. greater accessibility of cigarettes (40%). For those who switched from smoking to vaping (n = 178), the predominant motivations included 1. having friends or family who vape (40%) and 2. the perception that vaping was healthier (36%). Among multi-inhalant users (n = 223), key factors driving this behavior included 1. increased enjoyment (47%) and 2. greater variety (42%). Our findings imply that there is no single, dominant reason driving the initiation of combustible use or switching from one inhalant to another. Further, tobacco users are receiving mixed messaging, leading many to believe that combustible tobacco is the healthier option. Public health interventions are needed to prevent the initiation of e-cigarette vaping by youth and to educate the public about the health effects of tobacco products.

## 1. Introduction

When e-cigarettes were introduced to the United States in 2007, they were marketed as a safe alternative to combustible cigarettes. Tobacco use in adolescents had been constantly decreasing, but with the introduction to the market of the fruity, sweet, and minty flavored e-cigarettes, tobacco use rapidly increased [1], with use by middle school (ages 10–14), high school (ages 14–18), and college students (ages 19–23) surpassing that among adults [2]. For example, 18.6% of college students between the ages of 19 and 22 have reported using an e-cigarette within the past 30 days [3]. Amidst this surge, studies have deliberated on whether e-cigarette use among adolescents poses a risk factor for subsequent cigarette smoking—a phenomenon often termed the gateway effect [4,5,6,7]. According to this theory, e-cigarettes may function as a nicotine starter, leading to nicotine dependence and the subsequent use of other tobacco products [8]. Meta-analyses in 2017 and 2021 have confirmed that the initiation of nicotine e-cigarettes by adolescents increases the subsequent use of combustible tobacco [1,9]. Data thus far also suggest that the initiation of e-cigarettes leads to the multi-inhalant use of vapes with combustible tobacco and cannabis [10], a pattern of inhalant use that has more adverse effects than either inhalant alone [3,8]. Most notably, it remains unknown why never-smokers who vape e-cigarettes begin smoking conventional tobacco.

In this study, we assess the driving forces underlying the initiation of combustible tobacco use by e-cigarette vapers. We posit that nicotine addiction plays a central role in switching from vapes to cigarettes and transitions to multi-inhalant use, and that combustible tobacco generates a stronger nicotine ‘high’ leading to greater pleasure with this type of inhalant. We hypothesize that other factors also contribute, such as the greater accessibility of cigarettes. Furthermore, we explore other patterns of inhalant usage, including transitions from cigarette smoking to vaping nicotine, multi-inhalant use, and the cessation of both habits. We anticipate that individuals may opt for vaping over smoking due to perceived health benefits, while those engaging in multi-inhalant use may seek heightened nicotine intake. Finally, we hypothesize that the primary driver for quitting both e-cigarettes and combustible cigarettes is health benefits.

By examining inhalant use patterns and motivations underlying the choices tobacco users are making, we hope to obtain a deeper understanding of the complex dynamics surrounding e-cigarette and cigarette use, shedding light on potential avenues for interventions, public health outreach, and harm reduction strategies.

## 2. Materials and Methods

### 2.1. Survey Design and Distribution

A 33-item questionnaire was designed in Survey Monkey, with inhalant questions based on our published University of California San Diego (UCSD) Inhalant Questionnaire [10,11,12] and the University of Colorado Anschutz 2021 Healthy Kids Survey [13]. Novel questions probing behavioral drivers underlying switching and the addition of combustible tobacco in vapers were also included. UCSD institutional review board (IRB 160204) approval was obtained, and all survey takers gave consent (>18 years of age) or assent (<18 years of age). The focused survey was posted on multiple social media platforms, including Facebook, Craigslist, and Instagram. To avoid and reduce bot responses, Twitter and Reddit were not used. All survey takers were entered into a random drawing for a USD 100 gift card.

### 2.2. Assessment for ‘Bots’

Individuals completed two bot-detection questions within the survey. A total of 1270 subjects, composed of nonsmokers, e-cigarette users, smokers, and multi-inhalant users, completed our nationwide online survey between October of 2023 to February of 2024, with 864 non-bot survey responses identified (Figure 1). The first bot detection question presented participants with a photo displaying a road number and name and prompted them to input the road name into a designated text box. Responses lacking either the road number or name were flagged and subsequently excluded. The second bot detection approach employed questions about participants’ ages using two distinct formats: written and visual. Responses showing discrepancies exceeding a difference of 3 years or failing to match between formats were flagged and excluded from analysis. Subjects who reported the use of vaping products beginning >15 years ago were excluded from analysis. A total of 864 complete, non-bot, survey responses were identified, of which 609 were users of inhalants and were included for analysis (Figure 1).

### 2.3. Analysis

Raw data were exported into Microsoft Excel. Organized data were graphed and analyzed with GraphPad Prism (version 10.3.0; San Diego, CA, USA). Comparisons between two groups were analyzed with non-parametric *t*-tests (Mann–Whitney). Analyses of three groups utilized one-way ANOVA with Dunn’s multiple comparisons test. The mapping of IP addresses to identify countries of origin was conducted using ipinfo.io, and map creation was performed in both ipinfo.io and ip2location.com.

## 3. Results

### 3.1. Demographic Characteristics of the Survey Respondents

The survey respondents were 29 years old on average (range of 14 to 80) and the majority were male (59% male and 7% transgender male; Table 1). The subjects were predominantly located in the United States (n = 714; Figure 2A), with respondents additionally from Canada (n = 8), Sweden and Venezuela (n = 3 each), and South Korea, Singapore, Hong Kong, China, Ukraine, France, the Dominican Republic, Nigeria, and Mayotte (n = 1 each; Figure 2B). Users of vaping devices reported an initiation of use between ages 6 and 72 (median 19 years of age) while users of combustible cigarettes initiated earlier, between ages 6 and 78 (median 18 years of age; *p* < 0.0001, 95% CI 2.0 to 3.0; Figure 3A). At the time of the survey, users of e-cigarettes had a mean duration of use of 7.5 years (range 0–15, 95% CI 7.13 to 7.85) while users of combustible cigarettes had a longer duration of use of 11.3 years (range 0–48, 95% CI 10.75 to 11.9; *p* < 0.0001, 95% CI 2.0 to 4.0; Figure 3B).

For subjects who reported vaping product and combustible tobacco use, 57% reported smoking first for an average of 4.1 years prior to adding vaping devices (range of 1–24 years), 32% reported initiating both smoking and vaping at the same time (within 12 months), and 11% reported initiating vaping first for an average of 3 years before initiating the use of combustibles (range of 1–10 years). Forty-four percent of inhalant users also reported the concomitant use of cannabis, with 84 reporting use of combustible forms alone, 112 of vaped forms alone, and 73 of edible forms alone. In addition, twenty-three reported the use of both combustible and vaped forms of cannabis; seventeen used combustible, vape, and edible forms; eleven used both vape and edible forms; and six used both combustibles and edibles.

E-cigarette users reported vaping an average of 18 days out of the last 30, with a range of 2–30 days (95% CI 17.7 to 19.1). Combustible tobacco users reported a similar frequency of use, smoking 19 days of the last 30, with a range of 1–30 (95% CI 18.2 to 19.8; *p* = 0.17, 95% CI 0.0 to 2.0; Figure 3C). Survey respondents indicated that vaping and smoking were equally enjoyable on a scale of 1 to 100 (67.8 versus 68.1, respectively; *p* = 0.90; Figure 3D). Subjects reported that combustible cigarettes were easier to obtain than vaping products on a scale of 1 to 10 (7.2 versus 6.8, respectively, *p* < 0.01, 95% CI 0.0 to 1.0; Figure 3E). Subjects reported knowing equal numbers of people who vape or smoke (27.3 with 95% CI 25.7 to 29.0 versus 27.6 with 95% CI 26.0 to 29.3, respectively) but reported knowing fewer numbers of people who use cannabis (16.9, CI 15.5 to 18.4, *p* < 0.0001; Figure 3F).

### 3.2. Factors Contributing to Switching from Vaping Nicotine (Using E-Cigarettes) to Smoking Cigarettes

Out of 104 inhalant users who reported switching from vaping to smoking, 44% stated that they believed smoking was healthier than vaping (Figure 4A). This was surprising but not entirely unexpected, as there have been multiple anti-vaping campaigns over the last several years, such as those by the Truth Initiative, the Real Cost Campaign, and the Empower Vape-Free Youth Campaign. Forty percent of those who switched from vaping to smoking noted that cigarettes were more accessible, and 38% mentioned having friends or family who smoke cigarettes as a factor in their decision. Among participants who provided only one reason for their switch, believing that smoking is healthier than vaping was the most common choice. None of the participants selected reasons such as social pressure, changing from sweet to harsher forms, the normalization of smoking-related behaviors by e-cigarette vaping, or other miscellaneous reasons (Figure 4A).

### 3.3. Factors Contributing to Switching from Smoking Cigarettes to Vaping Nicotine

Among the 178 inhalant users who switched from smoking to vaping, 40% highlighted having friends or family who vape as a driver, 36% expressed the belief that vaping is healthier than smoking, and 33% cited the better flavor of vapes as a reason for their switch (Figure 4B). This was an anticipated result, as e-cigarette flavors as a driver of initiation by cigarette smokers have been identified in other studies [14]. Having a friend or family member who vapes was cited as a top reason among participants who selected only one reason for their switch (Figure 4B). This is a known driver of e-cigarette vaping initiation in never-smoker/never-vapers [15], and these data confirm that it plays a role in combustible smokers as well.

### 3.4. Multi-Inhalant Use of Nicotine Vapes and Combustible Cigarettes

Among 223 participants who identified as multi-inhalant users, 47% (n = 105) highlighted increased enjoyment as the reason for multi-inhalant use, while 42% (n = 93) expressed a desire for more variety as a driver of this behavior (Figure 4C). Additionally, 35% (n = 78) reported having people around them who use multiple inhalants. Furthermore, 29% (n = 64) attributed their multi-inhalant usage to obtaining more nicotine, and 29% (n = 64) said they do it to increase peer acceptance or respect. Moreover, 28% (n = 62) mentioned that they smoke and vape because of the flavor, and 27% (n = 60) said they do it for stress relief. Participants who picked only one reason for multi-inhalant use chose 1. increased enjoyment, 2. increased peer acceptance or respect, or 3. flavor. No participants mentioned other reasons for their multi-inhalant use (Figure 4C).

### 3.5. Quitters of Tobacco Use

Out of the 93 participants who quit smoking and/or vaping, 66% (n = 61) attributed quitting to have better health, while 47% (n = 44) mentioned having a friend, family member, or doctor who encouraged them to quit as a contributing factor (Figure 4D). Additionally, 43% (n = 40) stated that it was to improve the environment, and 28% (n = 26) expressed that smoking and vaping are expensive. The top choice among participants who picked only one reason for quitting was better health.

## 4. Discussion

In this study, we explored the reasons underlying the initiation of combustible tobacco use by e-cigarette vapers. Our findings reveal a nuanced picture, with multiple reasons contributing to initiating one inhalant in the setting of existing inhalant use, as well as switching between inhalants (infographic). Specifically, nicotine addiction did not emerge as the dominant factor according to respondents’ perceptions. Instead, participants cited a variety of motivations, including beliefs that smoking is healthier than vaping, the influence of social networks, the lower cost of cigarettes, and taste preferences. This multifaceted explanation challenges previous assumptions that nicotine addiction is the sole driver of transitioning from vaping to smoking.

An unexpected finding was the perception among some participants that smoking is healthier than vaping. This finding contradicts the prevailing literature, which broadly has determined that vaping is a less harmful alternative to smoking [16,17]. This finding is notable because it highlights a public health need for clear messaging about the known health effects of different forms of tobacco. While the public should be encouraged not to initiate e-cigarette vaping, as it does have adverse effects on health [11,18,19,20] and may lead to higher initiation of combustible tobacco [21,22,23], those who already use combustible cigarettes should be informed of the potential harm reduction of switching completely from smoking to vaping [12]. However, we are now finding that multi-inhalant use of both combustible tobacco and vaped nicotine has greater negative effects on health than either inhalant alone [13,19], and many smokers who add in e-cigarettes to reduce cigarette consumption end up as multi-inhalant users [24]. Thus, the dangers of multi-inhalant use need to be clearly conveyed to the public as well.

In addressing a second research question as to why cigarette smokers switch to vaping nicotine, vaping being perceived as a healthier alternative, better flavors, and a lack of offensive smell emerged as significant reasons. The last two factors are particularly understandable, as vapes are known to come in a multitude of flavors and lack the pungent smell of cigarette smoke. The fact that this population of cigarette smokers views e-cigarette vaping as a healthier alternative is promising, as it suggests that some advocacy efforts to share the differences between combustibles and vapes have been successful. However, as with switching from vaping to smoking, our results revealed a diverse set of motivations for making the switch from smoking to vaping, including influence from social circles. It has long been known that peer use of any substance makes it more likely that an individual will initiate use of that substance [15,25,26,27], and our data confirm the same to be true of e-cigarettes. Interestingly, reasons driving switching from smoking to vaping overlapped with those for transitioning from vaping to smoking, suggesting complex dynamics at play in individuals’ decisions regarding inhalant usage.

Multi-inhalant users of nicotine vapes and cigarettes were motivated by a desire for increased nicotine intake, as well as for enjoyment and variety. The prioritizing of increased nicotine intake by multi-inhalant users supports nicotine addiction as a driver of continued use of both combustible and vaped products. Smoking and vaping of tobacco were rated as similar in terms of how much users enjoy inhaling each (Figure 3D), which supports the continued use of both inhalants once someone has initiated both. If one of the inhalants conferred more pleasure, as combustible tobacco cigarettes were known to do relative to earlier generations of e-cigarettes that did not have as effective nicotine delivery as 4th generation e-devices [28], then we would anticipate more subjects becoming single-inhalant users.

We hypothesized that if one product was more difficult to obtain, that would drive inhalant users towards more use of the product that is easier to obtain. Or, in the case of users of the more difficult to obtain inhalant, this would drive them towards the initiation of the other inhalant. In this study, we found that combustible cigarettes were slightly easier to obtain (Figure 3E), which suggests that access may be playing at least a small role in the initiation of combustible use by e-cigarette vapers. However, the underlying nicotine addiction would still be considered the primary driver.

Both quitters of vaping and quitters of smoking cited concerns about their health; encouragement from friends, family, or healthcare providers; environmental considerations; and the cost of smoking/vaping as motivations for quitting. Additionally, eleven participants described their inhalant usage as occasional, particularly in social settings. While not fitting neatly into the categories of switching or quitting, these experiences offer insights into the context-specific nature of inhalant use and the role of social influences.

Limitations of this study include possible selection bias introduced through recruitment via social media. This is an inherent limitation of all social media-based survey studies. We attempted to limit selection bias by posting the survey across platforms, with a variety of advertisements, and by posting across days and at multiple times of day. Additionally, the incentive offered for survey participation may have influenced participant motivation. However, surveys without incentives have very low click-through and completion rates, necessitating this methodology. Due to the need for brevity, because longer surveys have lower click-through and completion rates [29], sociodemographic data beyond age, gender, race, and ethnicity were not obtained. Further, in-depth details about inhalant use, such as e-device type, e-cigarette flavors, and nicotine type and concentration, were also not obtained. Future research should explore inhalant usage patterns in diverse populations and use detailed inhalant questions to rigorously define the types of inhalant use. Further, future studies should include nicotine addiction assessments to better define the role it plays in multi-inhalant use. Because of the limitations of anonymous, social media-based studies, future studies most optimally would utilize in-person interviews.

## 5. Conclusions

There are complex reasons underlying inhalant use patterns, with a diverse array of motivations for initiating, switching, multi-inhalant use, and quitting. Although our findings from this pilot study are exploratory, they underscore the need for targeted interventions that address individual preferences, social influences, and perceptions of harm to effectively promote smoking and vaping cessation and harm reduction. Specifically, sharing these findings with organizations highly experienced in communicating with the public about the harms of smoking, such as Tobacco-Free Kids, the Truth campaign, and the Tips campaign, to show them where important messaging is needed, would likely lead to the greatest impact on public health. We would recommend specific messaging, such as 1. smoking tobacco is more harmful to your health than vaping, 2. combining smoking with vaping has worse health effects than either inhalant alone, and 3. vaping adversely affects your health.

Further research, using rigorous methodology and detailed interviews, is warranted to explore these dynamics in more depth and inform evidence-based strategies for addressing the public health challenges associated with smoking and vaping. Greater regulation of e-cigarette devices is needed to prevent the initiation of vaping in adolescents and young adults and prevent transitioning to combustible use and multi-inhalant use, both of which are known to cause more adverse health effects than vaping alone.

## Figures and Tables

**Figure 1 ijerph-22-00800-f001:**
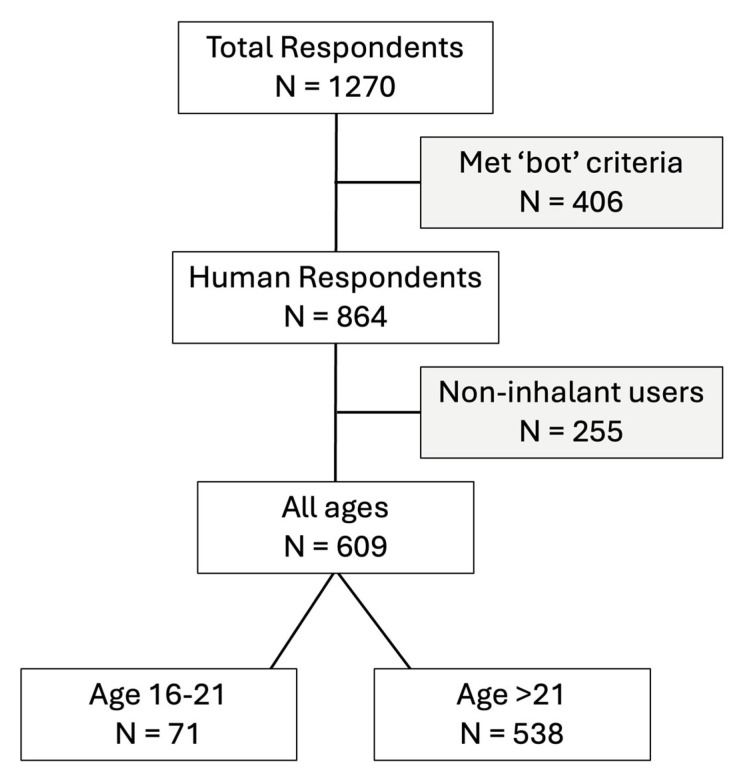
Breakdown of total survey respondents who met criteria for inclusion into the study. A ‘bot’ is a computer program designed to automate tasks on the internet.

**Figure 2 ijerph-22-00800-f002:**
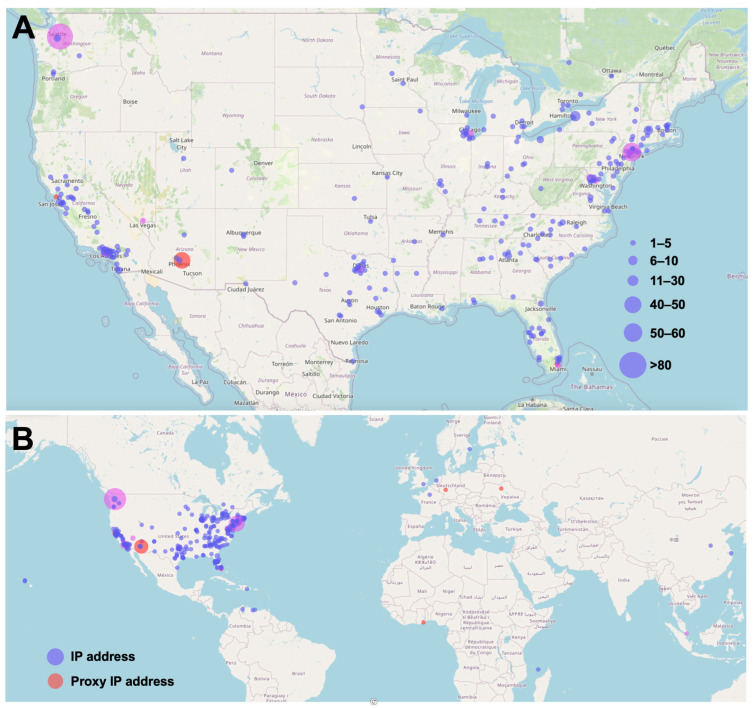
Location of survey respondents based on IP addresses. (**A**) The majority of survey respondents were located in the United States. (**B**) Subjects from 12 additional countries participated. Countries not shown did not have any survey respondents. Maps were created by inputting an IP address recorded by SurveyMonkey into map.ip2location.com. The number of survey respondents is indicated by the size of the pin. Blue pins represent internet protocol (IP) addresses, red indicates proxy IP addresses (survey responders computers routed through a proxy server, masking the actual location of the survey respondent), and pink and purple represent a mix.

**Figure 3 ijerph-22-00800-f003:**
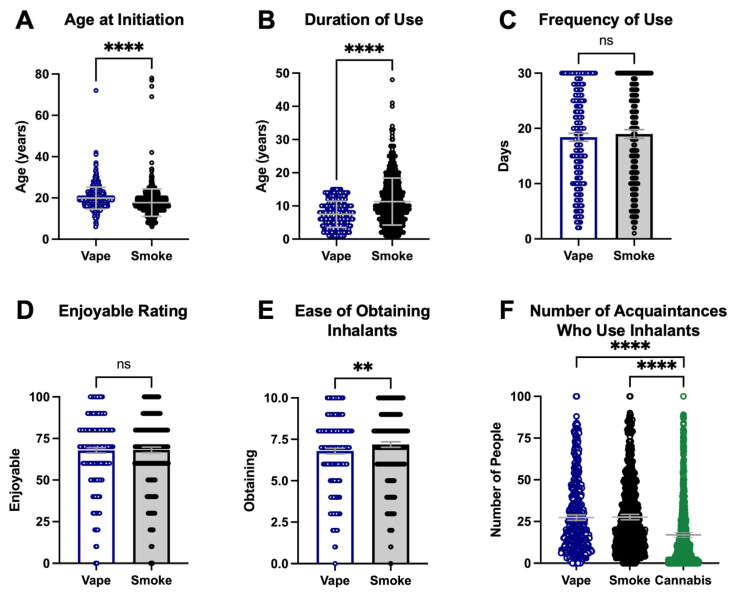
(**A**) Users of combustible tobacco reported an earlier age of initiation of smoking relative to users of vaping products. (**B**) Combustible tobacco users had longer durations of use than e-device users at the time of the survey. (**C**) The number of days of use over the last 30 days was similar between e-device vaping and combustible tobacco smoking groups. (**D**) Survey respondents rated vaping e-devices to be equally enjoyable to smoking combustible tobacco. (**E**) Subjects reported that it was easier to obtain combustible tobacco products. (**F**) Survey respondents reported knowing more people who vape and smoke than users of cannabis products. Error bars indicate the mean and 95% CI. ** *p* < 0.01, **** *p* < 0.0001.

**Figure 4 ijerph-22-00800-f004:**
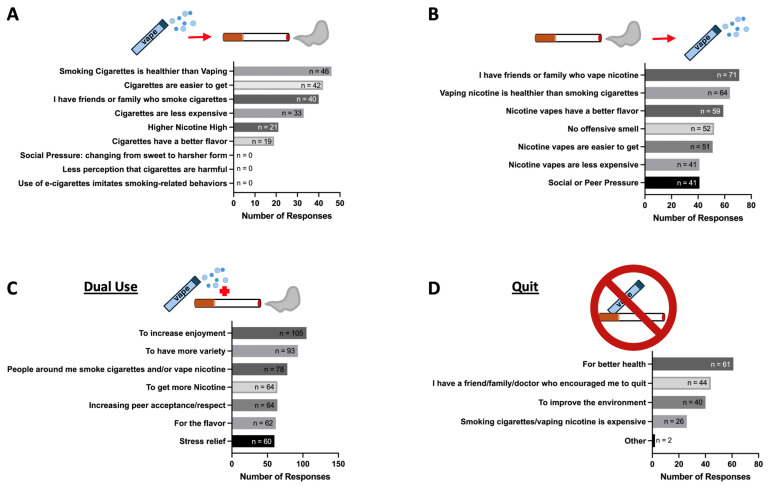
Factors contributing to switching from vaping to smoking, smoking to vaping, and multi-inhalant use. (**A**) The main drivers of switching from vaping to smoking include the belief that smoking is healthier than vaping, the ease of obtaining cigarettes, and influence from friends or family who smoke. Notably, none of the respondents cited social pressure, the transition from sweet to harsh forms, reduced perception of harm, or the imitation of smoking-related behaviors by e-cigarettes as reasons for switching from vaping to smoking. (**B**) The main drivers reported for transitioning from smoking to vaping were having friends or family who vape, the belief that vaping is healthier than smoking, and the perception that vapes have better flavor. (**C**) The primary drivers for the multi-inhalant use of vapes and combustibles include increased enjoyment, having more variety, and being influenced by people around who smoke and/or vape. (**D**) The main motivators behind quitting smoking and vaping were for better health; encouragement from friends, family, or doctors; and to improve the environment.

**Table 1 ijerph-22-00800-t001:** Demographic data for survey respondents.

	*All Human Respondents* *609 (100%)* *N (%)*	*Switched: Vaping to Smoking* *104 (17%)* *N (%)*	*Switched: Smoking to Vaping* *178 (29%)* *N (%)*	*Dual Use* *223 (37%)* *N (%)*	*Quit* *Tobacco* *93 (15%)* *N (%)*	*Other* *Inhalant Use* *11 (2%)* *N (%)*
Age						
16–21 years	71 (12%)	15 (2%)	17 (3%)	24 (4%)	14 (2%)	1 (0%)
>21 years	538 (88%)	89 (15%)	161 (26%)	199 (33%)	79 (13%)	10 (2%)
Gender						
Female	183 (30%)	37 (6%)	50 (8%)	65 (11%)	28 (5%)	3 (0%)
Male	358 (59%)	53 (9%)	108 (18%)	130 (21%)	60 (10%)	7 (1%)
Transgender Female	23 (4%)	5 (1%)	7 (1%)	9 (1%)	2 (0%)	0 (0%)
Transgender Male	41 (7%)	9 (1%)	10 (2%)	19 (3%)	3 (0%)	0 (0%)
Gender queer/non-binary	4 (1%)	0 (0%)	3 (0%)	0 (0%)	0 (0%)	1 (0%)
Ethnicity						
Hispanic or Latino	49 (8%)	3 (0%)	20 (3%)	16 (3%)	9 (1%)	1 (0%)
Race *						
Black or African American	35 (6%)	3 (0%)	10 (2%)	15 (2%)	7 (1%)	0 (0%)
White	479 (79%)	93 (15%)	139 (23%)	177 (29%)	62 (10%)	8 (1%)
Asian or Asian American	56 (9%)	9 (1%)	15 (2%)	21 (3%)	9 (1%)	2 (0%)
American Indian or Alaska Native	39 (6%)	3 (0%)	11 (2%)	14 (2%)	11 (2%)	0 (0%)
Native Hawaiian or other Pacific Islander	13 (2%)	3 (0%)	3 (0%)	4 (1%)	3 (0%)	0 (0%)

* Multiple participants identified as two or more races.

## Data Availability

The raw data supporting the conclusions of this article will be made available by the authors on request.

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
