# Peer review of "Perceptions of Health Risks and Accessibility: A Social Media-Based Pilot Study of Factors Influencing Use of Vaping and Combustible Tobacco Products"

_ijerph, 2025, doi:10.3390/ijerph22050800_

Round 1
Reviewer 1 Report
Comments and Suggestions for Authors
Data presentation: There is a serious error in Table 1 that requires correction and possible error in Figure 1 . In Table 1, the numbers (N) and percentages (%) are reversed in all table items. In Figure 1, the sum of respondents meeting the bot and not-bot criteria does not match the total number of respondents.
Discussion, in line 149, I suggest: Nicotine addiction did not emerge as the dominant factor according to respondents' perceptions.
Author Response
Thank you so much for reviewing our manuscript. We have revised the manuscript in response to reviewer comments and believe the edits have significantly improved the paper.
Comment 1: Data presentation: There is a serious error in Table 1 that requires correction and possible error in Figure 1 . In Table 1, the numbers (N) and percentages (%) are reversed in all table items. In Figure 1, the sum of respondents meeting the bot and not-bot criteria does not match the total number of respondents.
Response 1: Apologies for the swapping of the N and % in the table. We have corrected the swapping, and took the opportunity to make the table a bit clearer as well (added more labeling). We moved Figure 1 up into the Methods and added detail to the figure to more accurately and clearly show the number of respondents excluded due to bot detection and the number excluded because they were non-inhalant users. We also clarified the numbers in the text, and added more methodologic detail.
Comment 2: Discussion, in line 149, I suggest: Nicotine addiction did not emerge as the dominant factor according to respondents' perceptions.
Response 2: Thank you for this suggestion, we have incorporated it into the discussion.
Reviewer 2 Report
Comments and Suggestions for Authors
Although this paper is on an interesting topic the research undertaken with an opportunistic sample is not really adequate for publication in a highly ranked peer reviewed journal. Ideally we would need to know a great deal more about the people completing the research instruments to understand the results presented within the paper. The analysis undertaken is very much broad stroke rather than a penetrating more detailed analysis of the data collected and what it says bout the views of those contributing to this research. I am sceptical whether the study as reported here would justify publication within the journal- at minimum the authors would need to substantially strengthen the analysis of the data collecte.
Author Response
Comment 1: Although this paper is on an interesting topic the research undertaken with an opportunistic sample is not really adequate for publication in a highly ranked peer reviewed journal. Ideally, we would need to know a great deal more about the people completing the research instruments to understand the results presented within the paper.
Response 1: We agree with the reviewer that we would have preferred to conduct an in-person, longitudinal, large study to answer our questions. However, in the absence of practically any data published on this critical and unexplored topic, we do feel that this study can act as a starting point, hopefully to convince funding agencies to support larger more rigorous studies in this arena. One of the critical weaknesses of conducting social media based surveys on adolescents is the need to keep the surveys very short due to short attention spans and question fatigue in this population. This leads to an inability to ask the extensive questions we would prefer to ask, to better define the subjects completing the study. We have added more explanations about the study design and the limitations to the manuscript. Further, we added a new section within the Results and incorporated as much detail about the inhalant users as possible, based on the limited number of questions that were used within the survey instrument. This has led to the addition of: location of survey respondents (new Figure 2), age of initiation of smoking versus vaping, duration of smoking and vaping, frequency of inhalant use, cannabis use rates, ratings of enjoyability of smoking versus vaping, ease with which combustible versus vaping products can be obtained, and number of acquaintances who smoke, vape, or use cannabis products (new Figure 3).
Comment 2: The analysis undertaken is very much broad stroke rather than a penetrating more detailed analysis of the data collected and what it says about the views of those contributing to this research. I am skeptical whether the study as reported here would justify publication within the journal- at minimum the authors would need to substantially strengthen the analysis of the data collected.
Response 2: We have added two new figures, one with multiple panels of individual data points, with rigorous analyses conducted in GraphPad Prism. We have added detail in the methods section about all analyses used. Because of the inherent limitations of survey-based science, we have added more detail to the discussion section about what next steps are needed in the field, to answer the critical questions regarding inhalant use behaviors and the reasons driving these behaviors. We are hopeful that this work can act as a springboard for starting the larger research studies needed.
Reviewer 3 Report
Comments and Suggestions for Authors
The paper addresses a current and innovative topic, however, it is important to note the following:
- The objective ir not clearly stated: For example, it is not specified in the abstract; at the end of the introduction, it states "we assess the driving forces underlying initiation of combustible tobacco by e-cigarette vapers". At the beginning of the discussion, it states, "in this study, we explored reasons underlying transitions from vaping to smoking".
- The results should be structured according to the general objective and specific objectives. It is suggested that it would be important to establish an objective that involves statistical analysis, since the results are presented only in terms of frequencies and percentages.
- The title is inconsistent with the method, the results, or the discussion.
- In the discussion, only the results section is repeated. Therefore, it should be structured according to the development of an analysis of the results obtained in the research presented with respect to the results obtained in previous studies, identifying limitations and making proposals for future research. It is also important to clearly state the contribution of the study.
Author Response
Comment 1: The paper addresses a current and innovative topic, however, it is important to note the following: The objective ir not clearly stated: For example, it is not specified in the abstract; at the end of the introduction, it states "we assess the driving forces underlying initiation of combustible tobacco by e-cigarette vapers". At the beginning of the discussion, it states, "in this study, we explored reasons underlying transitions from vaping to smoking".
Response 1: Thank you for noticing the lack of consistency across the different sections of the manuscript. Because our original objective was to identify the primary factors driving the initiation of combustible tobacco by e-cigarette vapers, we have added that to the abstract and made the versions in the introduction and discussion parallel to that in the abstract.
Comment 2: The results should be structured according to the general objective and specific objectives. It is suggested that it would be important to establish an objective that involves statistical analysis, since the results are presented only in terms of frequencies and percentages.
Response 2: We have added 2 new figures, with multiple panels, and have added more detail about the human subjects who completed this survey. We have made multiple changes to the results section, which we hope have improved its structure and clarity.
Comment 3: The title is inconsistent with the method, the results, or the discussion.
Response 3: We have changed the title to “Perceptions of Health Risks and Accessibility: A Social Media-Based Pilot Study of Factors Influencing Use of Vaping and Combustible Tobacco Products”. We hope this is a better fit for this exploratory study utilizing an online survey tool.
Comment 4: In the discussion, only the results section is repeated. Therefore, it should be structured according to the development of an analysis of the results obtained in the research presented with respect to the results obtained in previous studies, identifying limitations and making proposals for future research. It is also important to clearly state the contribution of the study.
Response 4: We have deep-edited the discussion to make it more insightful, and clearly identify what research needs to be done next.
Round 2
Reviewer 2 Report
Comments and Suggestions for Authors
The manuscript has been improved following revision. The key finding in this paper to do with the misaprehension of the level of harms associated with vaping and smoking (explaining smoking initiation following vaping) is important but somewhat under-addressed within the paper. At minimum the authors should strengthen the attention given to this finding within the conclusion section of the paper. At present although there is a call for tighter regulation of e-cigarettes such a recommendation hardly addressed why vapers might perceive smoking to be less harmful than vaping and what one might need to initiate to rectify that error in harm perception.
Author Response
Comment 1: The manuscript has been improved following revision. The key finding in this paper to do with the misapprehension of the level of harms associated with vaping and smoking (explaining smoking initiation following vaping) is important but somewhat under-addressed within the paper. At minimum the authors should strengthen the attention given to this finding within the conclusion section of the paper. At present although there is a call for tighter regulation of e-cigarettes such a recommendation hardly addressed why vapers might perceive smoking to be less harmful than vaping and what one might need to initiate to rectify that error in harm perception.
Response 1: Excellent point. We need to communicate the relative harms and dangers of combustible tobacco versus vaping. A call to arms on the advocacy and community engagement fronts is what we would like to happen. In terms of exactly how to get the most important points across to the general population, we aren’t sure what would work best, but we would defer to the advocacy groups who have been highly impactful in this area: Truth campaign, Tobacco-Free Kids, and the Tips campaign. We had originally said “While the public should be encouraged not to initiate e-cigarette vaping, as it does have adverse effects on health[11, 18-20] and may lead to higher initiation of combustible tobacco[21-23], those who already use combustible cigarettes should be informed of the potential harm reduction of switching completely from smoking to vaping[12]” and we have added this to specifically cover how to better communicate the greater harms of smoking vs vaping with the public: “Specifically, sharing these findings with organizations highly experienced in communicating with the public about the harms of smoking, such as Tobacco Free Kids, the Truth campaign, and the Tips campaign, to show them where important messaging is needed, would likely lead to the greatest impact on public health. We would recommend specific messaging such as: 1. Smoking tobacco is more harmful to your health than vaping; 2. Combining smoking with vaping has worse health effects than either inhalant alone; and 3. Vaping adversely affects your health.”